# Oceans as a Source of Immunotherapy

**DOI:** 10.3390/md17050282

**Published:** 2019-05-10

**Authors:** Bilal Ahmad, Masaud Shah, Sangdun Choi

**Affiliations:** Department of Molecular Science and Technology, Ajou University, Suwon 16499, Korea; bilalpharma77@gmail.com (B.A.); masaudghalib@hotmail.com (M.S.)

**Keywords:** immunotherapy, inflammation, marine drug, marine flora, Toll-like receptor

## Abstract

Marine flora is taxonomically diverse, biologically active, and chemically unique. It is an excellent resource, which offers great opportunities for the discovery of new biopharmaceuticals such as immunomodulators and drugs targeting cancerous, inflammatory, microbial, and fungal diseases. The ability of some marine molecules to mediate specific inhibitory activities has been demonstrated in a range of cellular processes, including apoptosis, angiogenesis, and cell migration and adhesion. Immunomodulators have been shown to have significant therapeutic effects on immune-mediated diseases, but the search for safe and effective immunotherapies for other diseases such as sinusitis, atopic dermatitis, rheumatoid arthritis, asthma and allergies is ongoing. This review focuses on the marine-originated bioactive molecules with immunomodulatory potential, with a particular focus on the molecular mechanisms of specific agents with respect to their targets. It also addresses the commercial utilization of these compounds for possible drug improvement using metabolic engineering and genomics.

## 1. Introduction

Immune system dysfunction leads to the development of allergies, autoimmune and chronic inflammatory diseases, and cancers. Inflammation has been suggested to be the principal cause of chronic illnesses such as obesity, diabetes, cancer, rheumatoid arthritis (RA), neurodegenerative, and autoimmune diseases (ADs). The hallmark of autoimmunity is chronic inflammation that leads to the release of pro-inflammatory cytokines and other mediators, known as danger-associated molecular patterns (DAMPS), that activate pathogen recognition receptors (PRR) expressed by immune cells. Autoantibodies recognize these DAMPs and activate myeloid immune cells with an enhanced inflammatory response, leading to exacerbation of the condition. This self-perpetuating cycle continues, in order to assist with injury repair [1,2,3].

Recent estimates suggest that 7.6–9.4% of the world’s population is affected by immune-mediated diseases. Such diseases include inflammatory bowel disease (IBD), type 1 diabetes mellitus (TIDM), and RA. Women are up to ten times as likely to be sufferers as men [4]. ADs are among the ten leading causes of death for women, and affect them disproportionately in every age group up to 75 years of age [5]. According to the NIH report, around 23.5 million Americans have ADs, while the American Autoimmune Related Diseases Association (AARDA) puts this figure at 50 million. More than 80 different ADs have been identified and 40 additional diseases are suspected as being ADs [6,7]. The holy grail of immunotherapy is the reprograming of the immune system to maintain or restore homeostasis, and there is an urgent need to develop such drugs.

The search for novel immunomodulators is challenging, despite the existence of considerable amounts of demographic and epidemiological data about ADs. Questions about how autoimmunity is triggered and how self-tolerance is broken down remain to be fully answered. Nevertheless, progression in our understanding of the pathophysiology of ADs has led to the possibility of identifying new drug targets and new effective compounds with significant therapeutic potential. Several drug discovery and development programs are focused on the search for bioactive compounds obtained from natural sources. The study of nature’s enormous arsenal of new bioactive compounds and natural metabolites has historically led to immense benefits with respect to drug discovery [8]. The bioactivity of marine-derived natural products is significantly higher than that of compounds of terrestrial origin. For instance, in a National Cancer Institute preclinical cytotoxicity screen, approximately 1% of the marine samples tested showed anti-tumor potential in comparison with 0.1% of the terrestrial samples tested [9].

Oceans cover three quarters of the Earth’s surface, and contain the world’s greatest biodiversity, with at least 300,000 species of animals and plants described to date [10,11]. The diverse environments of oceanic zones provide a rich source of organisms. The hostile and competitive oceanic environment, with its high salt concentration and variations in hydrostatic pressure, produces microorganisms which are resistant to various kinds of stress, leading to the production of unique biomolecules. This enormous source of marine microorganisms has been exploited as a source of medicine since ancient times. The oceans are an exceptional storehouse of structurally and chemically novel bioactive compounds with unique biological features that are not generally found in terrestrial natural products. Over 60% of the active compounds of marketed formulations are natural products or their synthetic derivatives or mimics [12]. Nearly 3000 new compounds from marine sources have been identified over the last few years, and some have entered clinical trials [13]. Due to improved technologies for deep-sea sample collection and large-scale drug production through aquaculture and drug synthesis, the extent of this activity has been increasing steadily. Advanced engineering and the use of new technologies such as scuba diving techniques, remotely operated vehicles (ROVs) and manned submersibles have opened up the scientific exploration of the marine environment [14,15,16]. 

Marine compounds have been shown to have significant activity in antiviral, analgesic, antitumor, immunomodulatory, and anti-inflammatory assays [17,18]. Currently, six out of nine drugs in the market for cancer treatment are of marine origin, while several more are under clinical trials [18]. Undoubtedly, in the area of cancer, the marine metabolic arsenal plays a dominant role. 

## 2. Oceanic Sources of Immunotherapy

Immunotherapies induce, enhance or suppress an immune response to ameliorate a pathological condition. Immune response activation therapies are used in vaccines for conditions such as cancer, while immune response suppression therapies can be used to prevent graft rejection and treat autoimmune conditions and allergies. The metabolic products of microbes, phytoplankton, and zooplanktons obtained from marine environments offer a great diversity of physical and chemical attributes. Exploitation of marine microorganisms, which are the source of new genes, has led to the discovery of new drugs and targets. 

### 2.1. Oceanic Bacteria

Marine bacteria synthesize a wide range of valuable compounds with potential pharmaceutical applications [19,20]. Bacteria have yielded novel anti-inflammatory agents such as pseudopterosins, topsentins, scytonemin, and manoalide, anticancer and immunomodulatory agents including sarcodictylin eleutherobin, bryostatins and discodermolide, and antibiotics like marinone. Lactobacilli stimulate and modulate the mucosal immune system by reducing the production of pro-inflammatory cytokines through their actions on nuclear factor-*κ*B (NF-*κ*B) pathways. This effect is mediated by the production of anti-inflammatory cytokine interleukin-10 (IL-10) and host defense peptide *β*–defensin 2. The production of immunoglobulin A (IgA), dendritic cell maturation, and cell apoptosis is enhanced in response to short chain fatty acids [21]. Three diketopiperazine derivatives, cyclo(L-Pro-D-Val), cyclo(L-Pro-L-Tyr), and cyclo(L-pro-D-Leu), have been derived from two marine bacteria, *Bacillus* sp. HC001 and *Piscicoccus* sp. 12L081. These compounds show anti-inflammatory effects by suppressing polyphosphate mediated septic responses and hyper-permeability through the inhibition of p38 mitogen-activated protein kinases (MAPKs) activation. Downregulation of tumor necrosis factor (TNF-*α*), IL-6, NF-*κ*B and ERK1/2 was also observed after administration of these three compounds [22]. A novel exopolysaccharide (EPS) EPS1-T14, a water-soluble non-toxic exopolymer obtained from the marine bacterium *Bacillus licheniformis*, is able to stimulate an immune response. EPS1-T14 exhibits antiviral activity, as it inhibits the replication of HSV-2 in human peripheral blood mononuclear cells (hPBMCs). EPS1-T14 also stimulates the Th1 cell-mediated immune response [23]. Another EPS, TA-1, isolated from the thermophilic marine bacterium, *Thermus aquaticus,* is the strongest candidate for the EPS-binding receptor such as toll-like receptors (TLRs). TA-1 stimulates the release of the proinflammatory cytokines TNF-*α* and IL-6 from murine macrophages via a TLR-2 mediated pathway [24]. Prodigiosin (Figure 1, **1**) derived from marine bacteria such as *Pseudoalteromonas denitrificans, Vibro psychroerythru*s, and *Serratia marcescens,* has a strong inhibitory effect on many protozoan, fungal, and bacterial species, and induces apoptosis in cancer cell lines, as observed by the development of characteristic DNA laddering and apoptotic bodies [25,26]. Cycloprodigisin, a stable analog of prodigiosin (Figure 1, **1**) from *Pseudoalteromonas dentrificans* inhibits TNF-*α* induced NF-*κ*B activation, as determined through luciferase assay. This NF-*κ*B-inhibitory effect of cycloprodigiosin was retained under multiple stimuli in HeLa, U373, and COS7 cell lines [27,28,29]. Some representative immunomodulatory and anti-inflammatory chemical constituents isolated from marine bacteria are listed in Table 1. 

### 2.2. Cyanobacteria

Cyanobacteria are a fascinating and versatile, biologically important group of bacteria. Cyanobacteria unquestionably constitute one of the richest sources of novel and existing bioactive compounds, including toxins with a wide range of pharmaceutical applications. Cyanobacteria-derived bioactive molecules have a broad spectrum of activities, including antibacterial, antiviral, protease inhibition, anticancer and immunomodulatory activity (Table 2) [44]. Marine cyanobacterial lipopolysaccharide (LPS) has the potential to act as a TLR4 antagonist [45]. A cyanobacterial LPS (CyP), isolated from *Oscillatoria planktothrix* FP1 shows antagonism to TLR4 activation and exhibits protective effects in inflammatory conditions. CyP acts as a potential inhibitor of the LPS-induced inflammatory response in human and mouse dendritic cells, inhibiting both the MyD88-dependent and MyD88-independent TLR4 signaling pathways. CyP completely inhibits LPS-induced IL-1*β*, TNF-*α*, and IL-8 production [46,47]. CyP, when co-incubated with *Porphyromonas gingivalis* LPS (Pg-LPS) inhibited IL-1*β*, IL-8, and TNF-*α* expression more efficiently than LPS. CyP can modulate the Pg-LPS-induced pro-inflammatory response, by blocking TLR4-MD2, and also by preserving miR-146a expression [48]. Malyngamides, a class of compounds derived from the marine cyanobacterium *Lyngbya majuscula* has potent anti-inflammatory activity. One compound of this class, Malyngamide F (Figure 1, **2**) acetate, can inhibit the production of nitric oxide (NO) and other inflammatory biomarkers in RAW 264.7 cells. It selectively inhibits the MyD88-dependent pathway, because LPS stimulation decreases IL-1*β* and increases TNF-*α* transcription in MyD88 knockout mice via an MyD88-independent pathway [49]. Polysaccharides that have been extracted from *Arthrospira platensis* (*Spirulina*) have been reported to exhibit immune modulation. Immulina is one of these polysaccharides, and has been reported to decrease levels of TNF-*α* and IL-4 in FcεRI-activated RBL-2H3 cells [50,51]. Potent grassystatin A-C were obtained from the marine cyanobacterium *Lyngbya confervoides*. In response to exogenous antigen, grassystatin A (Figure 1, **3**) significantly reduces the proliferation of T cells. Grassystatin A (Figure 1, **3**) inhibit upregulation of IL-17 and interferon-*γ* (INF-*γ*) in response to antigen presentation [52]. Some representative immunomodulatory and anti-inflammatory chemical constituents isolated from marine cyanobacteria are listed in Table 2.

### 2.3. Sponges 

Sponges are currently the most important source of biologically active natural marine products and are considered to be a treasure trove of drugs [53,54]. Due to their lack of physical defenses, they have evolved a wide suite of defensive chemicals to deter predators [55]. New biomolecules discovered from marine sponges have strong immunosuppressive activities (Table 3). Didemnins, members of a depsipeptide class of compounds isolated from the Caribbean tunicate *Trididemnum solidum,* exhibit a variety of biological activities [56]. In particular, didemnin B (Figure 1, **4**) is characterized as immunosuppressive, and inhibits lymphocyte activation [57,58]. Target deconvolution studies, which aim to identify the molecular targets of active hits, have revealed that didemnin B (Figure 1, **4**) binds to the eukaryotic elongation factor 1*α* and palmitoyl-protein thioesterase 1. Large amounts of didemnin B (Figure 1, **4**) were taken up by proliferating cells, so this compound appears to be a promising drug for cancer treatment or the suppression of activation of the immune system [59].

*Dysidea* sp. has contributed significantly to biomolecule production [18,60,61]. Polyoxygenated sterols derived from *Dysidea* sp., have been shown to have strong selective immunosuppressive capability, blocking the interaction between IL-8 and its receptor [62]. Pateamine A (Figure 1, **5**) derived from *Mycale* sp., selectively inhibits the production of IL-2 in the T and B cells that produce the secondary immune response [63,64]. Discodermolide (Figure 1, **6**), a unique immunosuppressive and cytotoxic agent, is isolated from *Discodermia dissolute,* a deep water sponge [65]. An evaluation by the Longley group found that (+)-discodermolide has immunosuppressive properties at low concentrations, both in vivo and in vitro [66]. (+)-discodermolide was found to suppress the two-way mixed lymphocyte reaction in hPBMCs and murine splenocytes.

**Table 2 marinedrugs-17-00282-t002:** Marine cyanobacteria and their therapeutic chemical constituents.

Cyanobacteria Species	Chemical(s)	Immunomodulatory Activity	Ref(s)
*Oscillatoria planktothrix*	CyP	CyP modulates pro-inflammatory effect and inhibits TNF-*α*, IL-1*β* and IL-8.	[48]
*Lyngbya majuscula*	Microcolin-A (Peptides)	Suppresses murine splenocytes and inhibits LFA-1 and ICAM-1 mediated cell adhesion.	[49,67]
*Lyngbya sordida*	Malyngamide 2 (lipopeptide)	Inhibits production of NO in LPS-primed RAW 264.7 cells.	[68]
*Arthrospira platensis*	Immolina (Polysaccharide)	Reduces TNF-*α* and IL-4 levels in RBL-2H3 FcεRI-activated cells.	[50,51]
*Trichodesmium Erythraeum*	Aqueous extract	Anti-inflammatory effects in carrageenan-induced inflammation in rats.	[69]
*Lyngbya cf. confervoides*	Grassystatins A-C	Inhibits presentation of T cell antigen and expression of Cathepsin E, IL-17, and IFN- *γ*.	[52]

CyP, cyanobacterial LPS; ICAM-1, intercellular cell adhesion molecule-1; IFN, interferon; IL, interleukin; LFA-1, lymphocyte function-associated molecule-1; TNF, tumour necrosis factor.

**Table 3 marinedrugs-17-00282-t003:** Marine sponges and their therapeutic chemical constituents.

Sponge Species	Chemical(s)	Immunomodulatory Activity	Ref(s)
*Plakortis simplex*	Simplexides Glycolopids	Inhibits T cell proliferation and induces cytokines and chemokines in a CD1d-dependent manner.	[70,71]
*Dysidea* sp.	Dendroceratida & bolinaquinone (Polyoxygenated sterols)	Inhibits neutrophilic infiltration and IL-1, IL-8, PGE2, COX-2 expression in vivo.	[62,72]
*Petrosia contignata*	Contignasterol (Oxygenated sterol)	Inhibits histamine release in mast cells.	[73]
*Petrosia* sp.	Petrocortyne A (polyacetylenic alcohols)	Inhibits macrophages, reduces the production of TNF-*α* and the expression of phlogistic infiltration cell factors.	[27,74]
*Mycale* sp.	Pateamine (Thiazole macrolide)	Specifically targets translation initiation factors. Inhibits eIF4A-eIF4G association and promotes stable ternary complex formation between eIF4A and eIF4B. IL-2 inhibitor.	[75,76]
*Callyspongia* sp.	Callyspongidiol (Polyketide)	Dendritic cell activation with enhanced IL-4 and IL-10 production.	[77]
*Ianthella quadrangulata*	Iso-iantheran (Polyketide)	Has implication in tumor or autoimmune diseases. Ionotropic P2Y_11_ receptor activation.	[78]
*Xestospongia bergquisita*	Xestobergsterol (Polyhidroxylated steroid)	Inhibits the generation of IP3 and PLC activity and intracellular Ca^2 +^ mobilization.	[79]
*Clathria* *lissosclera*	Clathriols (Polyoxygenated steroids)	Inhibits superoxide production from neutrophils of hPBMCs.	[80]
*Hyritos sponge*	Heteronemin (Sesterterpene)	Inhibits TNF-*α* induced NF-*κ*B activation and induces caspase-dependent apoptosis in K562 cells.	[81]
*Xestospongia testudinaria*	Methanolic extract	Exhibits anti-inflammatory activity against carrageen-induced paw inflammation.	[82]
*Plakortis angulospiculatus*	Plakortide P	NO inhibition in LPS stimulated macrophages.	[83]
*Geodia cydonium*	Methanolic and Chloroform extraction	Reduces IL-8, CXCL10 and VEGF levels and increases IL-4 and IL-10 levels.	[84,85]
*Coscinoderma mathewsi*	Coscinolactams A-B (Terpenes) & suvanine	PGE2 and NO inhibition in RAW 264.7 cells stimulated by LPS.	[86]
*Lobophytum crassum*	Lobocrassin B	Inhibits LPS-induced BMDC activation by inhibiting TNF-*α* production.	[87]
*Petrosaspongia nigra*	Petrosaspongiolide	Inhibits chronic inflammation by lowering the production of eicosanoids and TNF-*α*.	[88]
*Hyrrios erecta*	Puupehedione, dipuupehedione, bispuupehenone	Exhibits cytotoxic and immunomodulatory potential against A-549 human cancer cell line.	[89]
*Gelliodes fibrosa*	Terpenes, steroids and lipids	Ethyl acetate extracts from *Gelliodes Fibrosa and Tedania anhelans* on in vivo carbon clearance tests showed a moderate immunostimulant effect.	[90]
*Ircinia variabilis*	Fasciculatin (Sesterterpenes)	Exhibits moderate cytotoxicity and no selectivity in the cancer cell lines.	[91]
*Dendrilla nigra*	Lipopolysaccharides & neolamellarins	*Dendrilla* exhibits enhanced phagocytosis against *Escherichia coli.* Neolamellarins inhibits HIF-1 activation and VEGF secretion in T47D cells.	[92,93]
*Theonella swinhoei*	Solomonsterol A, perthamides C & D (Peptides)	*Theonella* peptolides show mild immunosuppressive activity, inhibition of murine hind paw oedema.	[94,95]
*Discodermia spp*	Discodermolide (Polyhydroxylated lactone)	Inhibits murine T cell proliferation and causes cell cycle arrest in gap2 or mitosis phase of human and murine cell lines.	[96]
*Reniera spp*	Cyclic Tripeptide (Renieramide)	In preliminary tests renieramide showed immunomodulating activity.	[97]
*Trididemnum solidum*	Didemnins depsipeptides	Inhibits viral replication in vitro and *P388* leukemia in vivo.	
*Pseudoaxinyssa cantharella*	Girolline	Inhibits of IL-8, NF-*κ*B and AP-1 in macrophages derived from THP1. Reduction of IL-8 and IL-6 in primary mononuclear human cells.	[98]
*Callyspongia siphonella*	Callysterol (Sterol)	Potentially inhibits rat hind paw oedema, reduced release of TXB2 from LPS-activated rat brain microglia.	[99]
*Axinella verrucosa, Acanthella aurantica and Stylissa massa*	Alkaloids	Inhibits expression of NF-*κ*B and production of IL-8, IL-2 and TNF-*α*.	[100]
*Tedania ignis*	Tedanol (Diterpenoid)	Potent anti-inflammatory action to reduce carrageenan-induced mouse paw oedema. Strong inhibition of COX-2 and iNOS expressions.	[101,102]
*Haliclona* sp.	Halipeptins (Depsipeptide)	Strong anti-inflammatory activity, in vivo and in vitro.	[103]
*Cacospongia* *mollior*	Sesterterpenoid	Suppresses the production of LPS-induced PGE2.	[104]
*Fascaplysinopsis* *Bergquist* sp.	Fascaplysin (Indole alkaloid)	CDK 4 inhibitor, potential to elicit anti-neuroinflammatory or neuroprotective responses in neuroinflammatory disease models.	[105]
*Terpios* sp.	Terpioside B (Glycolipid)	Inhibits macrophage iNOS expression.	[106]

AP-1, activator protein; CDK, cyclin-dependent kinase 4; HIF-1, Hypoxia-inducible factor-1; IFN, interferon; IL, interleukin; iNKT, Natural killer T cells with an invariant T cell receptor alpha chain; iNOS, inducible nitric oxide synthase; NF-*κ*B, nuclear factor-*κ*B; PGE2, prostaglandin E2; PLC, phospholipase C; TNF, tumour necrosis factor; TXB2, thromboxane B2; VEGF, Vascular endothelial growth factor.

In several other non-lymphoid cell lines (+)-discodermolide exhibited antiproliferative effects by arresting the cell cycle at G2 and M phase due to microtubule network stabilization [107,108]. Petrosaspongiolide M (Figure 1, **7**) isolated from the Caledonian marine sponge *Petrosa spongia nigra* significantly inhibits chronic inflammation in rats and mice by diminishing eicosanoids and TNF-*α* production [88]. Petrosaspongiolide M (Figure 1, **7**) decreases the NF-*κ*B-DNA binding in response to zymosan in mouse peritoneal macrophages [109]. A marine sesterterpene, heteronemin (Figure 1, **8**), isolated from *Hyritos* sponge species has been found to affect cellular processes including cell cycle and apoptosis, and inhibits TNF-*α*-induced NF-*κ*B activation [81]. Methanolic extracts of *Xestospongia testudinaria*, the Red Sea marine sponge, prevent carrageenan-induced acute local inflammation in rats. Malondialdehyde and NO in inflamed rat paws was decreased by this extract, while glutathione, glutathione peroxidase, and catalase activities were increased. It appears to have antioxidant, anti-inflammatory, and immunomodulatory effects [82]. Some representative immunomodulatory and anti-inflammatory chemical constituents isolated from marine sponges are listed in Table 3. 

### 2.4. Algae

Marine algae are rich sources of vitamins, minerals, essential amino acids, lipids, fatty acids, dietary fiber, and polysaccharides [110,111,112,113,114]. Bioactivity studies of marine algae have revealed numerous health-promoting effects including anticoagulant, antibacterial, anti-hypolipidemic, anti-hypertensive, antioxidant, anticancer, and immunomodulatory activities [115]. Fucoidans (Figure 1, **9**) from the brown algae *Laminaria cichorioides*, *Laminaria japonica* and *Fucus evanescens* specifically interact with TLRs in vitro, causing the activation of NF-*κ*B via the MyD88 and TRIF-signaling pathways [116]. Brown algae-derived fucoidans activate genes that are responsible for cytokine synthesis, exhibiting pronounced immunotropic activity *ex vivo* and promoting defense against various pathogens [117]. Sugariura *et al.* showed that a diet including dried *Eisenia arborea* powder reduced serum IgE levels and shifted the Th1/Th2 balance by suppressing the release of Th2-type cytokines IL-4 and IL-10 and enhancing the expression of Th1 and IFN-*γ* in rat spleen and mesenteric lymph node-derived lymphocytes [118]. The red alga *Gracilaria verrucosa* has anticancer and antioxidant properties. The two enone fatty acids (E)-9-Oxooctadec-10-enoic-acid (Figure 1, **10**) and (E)-10-Oxooctadec-8-enoic-acid (Figure 1, **11**) isolated from *Gracilaria verrucosa* inhibit the production of inflammatory biomarkers including NO, IL-6 and TNF-*α* by suppressing the nuclear translocation of NF-*κ*B and phosphorylation of STAT1 in LPS-stimulated RAW 264.7 cells [119]. Lectins from marine algae *Solieria filiformis, Caulerpa cupressoides* and *Pterocladiella capillacea* demonstrate anti-inflammatory effects by enhancing IL-10 and IL-6 formation without affecting the IFN-*γ* and IL-12 production in murine splenocytes [120]. Some representative immunomodulatory and anti-inflammatory chemical constituents isolated from marine algae are listed in Table 4. 

### 2.5. Marine Fungi 

Recent developments in marine mycology have led to a large amount of research into natural products from substrate-insulated fungi in various marine habitats [121]. The discovery rate of novel marine-derived natural products from fungi increased exponentially over the period from 1970 to 2010 [121]. Cyclosporins (Figure 1, **12**) are produced by species of fungi including *Tolypocladium inflatum gams* [122], *Neocosmospora vasinfecta, Verticillium spp. and Microdochium nivale* [123,124]. Owing to their potent immune-modulation properties, cyclosporins (Figure 1, **12**) are used in patients with organ transplants. The agent specifically binds to cyclophilin expressed in T lymphocytes. The production of IL-2, IL-3, IL-4, granulocyte colony-stimulating factor (G-CSF) and TNF-*α* is reduced by cyclosporine in T-lymphocytes [125,126]. Sirolimus (Figure 1, **13**) (Rapamycin) a macrocyclic lactone immunosuppressive drug was also derived from the fungus *Streptomyces hygroscopicus* [127]. It binds to FK-bound protein 12 and serine threonine kinase, mTOR, inhibiting the transduction of IL-2R and other cytokine signals relevant to allograft rejection [128,129]. 

**Table 4 marinedrugs-17-00282-t004:** Marine algal flora and their therapeutic chemical constituents.

Algal Species	Chemical	Immunomodulatory Activity	Ref(s)
*Eisenia arborea*	Phlorotannin	Inhibits IgE and exhibits anti-degranulation effects; changes Th1/Th2 balance in Brown Norway rat strain.	[118]
*Endarachne binghamiae*	Polysaccharides (Sodium alginate, alginic)	Stimulates concentration-dependent proliferation of T cells and significant induction of the production of TNF-*α* and nitric oxide in macrophages and IFN-*γ* in T cells.	[130]
*Caulerpa cupressoides, Pterocladiella capillacea and Solieria demonstrate*	Lectins	Improves the IL-10 induction and induces the immune response of Th2 in mouse splenocytes.	[120]
*Gracilaria verrucosa*	Enone fatty acids	Inhibits the production of NO, TNF-*α*, and IL-6 inflammatory biomarkers.	[119]
*Sargassum ilicifolium*	Terpenes, steroids and lipids	Demonstrate chemotactic, phagocytic and intracellular killing of human neutrophils, and show a significant immunostimulatory effect in vivo.	[90,131]
*Laminaria japonica*	Laminarin oligosaccharides & polysaccharides	Apoptotic cell death protein was significantly reduced by laminarin oligosaccharides.	[132]
*Nannochloropsis oceanica*	Ethanol extract	Inhibits NO generation and downregulates NF-*κ*B and *β*-secretase activities in BV-2 cells.	[133]
*Monostroma nitidum*	Sulfated polysaccharides	RAW 264.7 cells were stimulated by polysaccharides, which produced considerable NO, and PGE2 induces strong immunomodulation.	[134]
*Hijikia fusiforme*	Polysaccharides	Enhanced activity for the proliferative response of spleen cells in endotoxin nonrespondent C3H / HeJ mice.	[135]
*Gyrodinium impudicum*	Polysaccharides	*Gyrodinium impudicum* show immunostimulatory effects and enhance the tumoricidal activities of macrophages and NK cells in vivo.	[136]
*Ulva fasciata*	Lipopolysaccharides	*Ulva* in the diet significantly increases defense factors such as haemogram, agglutination index, phagocytic rate, bacterial clearance and serum bactericidal activity.	[92]
*Sargassum thunbergii*	Fucoidan	Fucoidan enhances phagocytosis and macrophage chemiluminescence.	[137]
*Meristotheca papulosa*	Polysaccharides	Extracts of *M. papulosa* significantly stimulated the proliferation of human lymphocytes.	[138]
*Focellatus*	Carrageenan	λ-carrageenan showed antitumor activity and lymphocyte activation in mice transplanted tumor.	[139]
*Chlorella stigmatophora*	Polysaccharides	*Chlorella stigmatophora* extract shows anti-inflammatory effect in paw oedema test and immunomodulatory effects in delayed hypersensitivity test.	[140]
*Spirulina fusiformis*	Polysaccharides & *β* -carotene	*Spirulina fusiformis* suppresses adjuvant-induced arthritis in mice.	[141,142]
*Ceratodictyon spongiosum*	trans-ceratospongamide (Peptide)	Potent inhibition of sPLA2 expression in an anti-inflammatory cell model.	[143]
*Eisenia bicyclis*	Phlorotannins Dieckol, Eckol	Inhibits LPS-induced NO production, iNOS and COX-2 protein levels and t-BHP-induced ROS generation in RAW 264.7 cells.	[144,145]
*Eckolonia cava*	Fucodiphloroethol	Degranulation in RBL-2H3 cells induced by IgE.	[146,147]
*Rhipocephalus phoenix*	Rhipocephalin (Sesquiterpene)	Bee venom sPLA2 inhibitory activity.	[148]
*Crypthecodinium cohnii*	Exopolysaccharide EPCP1-2	Regulates the expression of TLR-4, MAPK and NF-*κ*B signaling pathways	[149]
*Gyrodinium impudium*	Sulphated polysaccharide P-KG103.	Activates NO production in a JNK-dependent manner and stimulates cytokines IL-1, IL-6, and TNF-*α* production in macrophages.	[136,150]
*Ishige okamurae*	Diphlorethohydroxycarmal-ol (Phlorotannin)	Inhibits the IL-6 production and expression of NF-*κ*B in murine macrophage RAW 264.7 cells.	[151]
*Fucus distichus*	Phlorotannin subfraction	Reduces TNF-*α*, IL-10, MCP-1 and COX-2 expression.	[152]
*Dinoflagellates* *(Protoceratium reticulatum, Lingulodinium polyedrum, Gonyaulax spinifera)*	Yessotoxin (Polyketide)	Inhibits macrophage phagocytosis and TNF-*α*, MIP-1*α* & MIP-2 expression.	[153,154]
*Laurencia claviformis, Laurencia filiformis, Laurencia tasmanica,* *Laurencia undulata*	Pacifenol (Terpenoid)	Anti-inflammatory activity, reduces the production of leukotriene B4 (LTB4) and thromboxane B2 (TXB2).	[155,156]
*Stypopodium flabelliforme*	Epitaondiol (Terpenoid)	Anti-inflammatory effects, inhibits the release and modulation of the COX pathway eicosanoids (LTB4 and TXB2).	[157,158]
*Lobophora variegata*	Lobophorins (Macrolides)	Anti-inflammatory properties.	[159]
*Cymopolia barbata*	Bromohydroquinones cymopol and cyclocymopol	Bee venom sPLA2 inhibitory activity.	[160]
*Stypoposium flabelliforme*	Meroterpene epitaondiol	Potent anti-inflammatory agent with strong activity on TPA induced ear oedema in mice and human neutrophils.	[161]
*Vidilia obtusaloba*	Bromophenols vidalols	Bee venom sPLA2 inhibitory activity.	[162]

COX, cyclooxygenase; IFN, interferon; IL, interleukin; JNK, c-Jun NH2-terminal kinase; IgE, immunoglobulin E; iNOS, inducible nitric oxide synthase; LKB4; leukotriene B4; LPS, lipopolysaccharide; MAPK; mitogen-activated protein kinase; MCP-1, monocyte chemoattractive protein-1; MIP, macrophage inflammatory protein; NF-*k*B, nuclear factor-*κ*B; NO, nitric oxide; PGE2, prostaglandin E2; PLA2, phospholipase A; t-BHP, tert-butylhydroperoxide; TLR, toll-like receptor; TNF, tumour necrosis factor; TPA, 12-O-tetradecanoylphorbol13-acetate; TXB2, thromboxane B2.

Semivioxanthin (Figure 1, **14**) from marine derived fungi are found to regulate the production of TNF-*α* and upregulate the expression of MHC II, CD80, and CD86 [163]. Brevicompanine E (Figure 1, **15**) isolated from the oceanic fungus *Penicillium* sp. is potentially useful for modulating neuroinflammation by attenuating NF-*κ*B and activator protein-1 (AP-1) activity in LPS-induced microglia [164]. Brevicompanine E (Figure 1, **15**) inhibits LPS-induced I*κ*B*α* degradation and NF-*κ*B nuclear translocation, and represses phosphorylation of c-Jun NH2-terminal kinase (JNK) and Akt (serine/threonine-specific protein kinase) [164]. Azonazine (Figure 1, **16**) hexacyclic dipeptide obtained from the Hawaiian marine sedimentary fungus *Aspergillus insulicola* exhibits anti-inflammatory activity by inhibiting the production of NF-*κ*B [165]. Three isocoumarins, (Figure 1, **17**) dichlorodiaportintone, desmethyldichlorodiaportintone, and desmethyldichlorodiaportinol, from the marine mangrove endophytic fungus, *Ascomycota* sp. CYSK-4, produce anti-inflammatory activity by inhibiting LPS-induced NO production in RAW 264.7 cells [166]. An anthraquinone derivative, questinol (Figure 1, **18**), isolated from the fungus *Eurotium amstelodami* exhibits an anti-inflammatory effect by significantly inhibiting prostaglandin E2 (PGE2) and NO production in LPS-stimulated RAW 264.7 cells. The production of pro-inflammatory cytokines, including IL-1, IL-6 and TNF-*α* is inhibited and inducible nitric oxide synthase (iNOS) expression levels suppressed in a dose-dependent manner [167]. Some representative immunomodulatory and anti-inflammatory chemical constituents isolated from marine fungi are listed in Table 5. 

Apart from their economic importance, fungi have been utilized as food, usually collected from their fruiting bodies, mushrooms. Some mushrooms can stimulate the immune system, modulate cellular and humoral immunity, and potentiate anti-tumorigenic activity, and potentially rejuvenate immune systems weakened by the chemotherapy and radiotherapy used for cancer treatment. This ability of mushrooms therefore qualifies them as candidates for immunotherapy in cancer and other diseases [168]. 

### 2.6. Mangroves and Other Higher Plants

Partially submerged in the ocean, mangroves form a tangled network of above-ground roots, which creates a unique and complex habitat for all sorts of marine life. Mangroves have long been used in fisher-folk medicine to treat disease [169,170]. Some mangroves, like *Rhizophora mangle* and *R. mucronata,* have been screened for their anti-ulcer, anti-viral, and anti-inflammatory activities [171,172,173]. Leaf extract from *Rhizophora apiculata* has been shown to inhibit HIV-1 or HIV-2 or SIV viruses in various cell cultures [174]. Extract of *Rhizophora apiculata* has shown anti-inflammatory and anti-tumor activity against B16F10 melanoma cells in BALB/c mice. *Rhizophora apiculata* substantially reduces acute inflammation in mice induced by carrageenan, as well as inflammatory oedema induced by formalin [175]. Extract of rhizome from *Acorus calamus* inhibited the growth of many human and mouse cell lines. In hPBMCs the production of IL-2, NO, and TNF-*α* was inhibited, IFN-*γ* and cell-surface markers CD16 and HLA-DR were not affected, but CD25 was downregulated [176]. Some representative immunomodulatory and anti-inflammatory chemical constituents isolated from marine mangroves are listed in Table 6. 

### 2.7. Marine Animals and Others

Entire marine animals, and their parts, contribute to the triggering of several biomedical mechanisms involved in inflammatory/allergic cascades [177]. An extract from the Caribbean Gorgonian *Pseudopterogorgia elisabethae* shows anti-inflammatory activity due to the presence of unusual diterpene glycosides, and is now used in cosmetic skin products as an anti-allergic factor [178]. Stichodactyla toxin (ShK)-186, a peptide toxin from sea anemones, blocks Kv1.3 potassium channels with a high degree of specificity. Kv1.3 potassium channels play a critical role in regulating the function of effector-memory T cells and class-switched memory B cells that are implicated in ADs [179]. Whole-body extracts of the marine prawn *Nematopaleamon tenuipes* (PEP), two gastropods, *Euchelus asper* (EAE) and *Hemifusus pulgilinus* (HPE), produced immunosuppression on Swiss albino mice in a concentration dependent manner [180]. An *α*-d-Glucan called MP-A, isolated from *Mytilus coruscus* (hard-shelled mussel), has shown anti-inflammatory activity in THP-1 human macrophage cells. MP-A suppresses LPS-induced TNF-*α*, NO, and PEG2 production via the TLR4 pathway [181]. Fatty acid extract from the tunicate *Halocynthia aurantium* significantly and dose-dependently increases NO and PGE2 production in RAW264.7 cells, producing immune enhancement without cytotoxicity. These fatty acids also regulate the transcription of immune-associated genes, including iNOS, IL-1*β*, IL-6, COX-2, and TNF-*α* [182]. Some representative immunomodulatory and anti-inflammatory chemical constituents isolated from marine creatures are listed in Table 6. 

**Table 5 marinedrugs-17-00282-t005:** Marine fungi and their therapeutic chemical constituents.

Marine Fungi	Chemical(s)	Immunomodulatory Activity	Ref(s)
*Neocosmospora vasinfecta*	cyclosporine	Calcineurine complex inhibition with cyclophilines.	[183]
*Streptomyces hygroscopicus*	Sirolimus macrocyclic lactone	Inhibits IL-2R signal transduction and other cytokine signals.	[127,128]
CTD-13C	Semivioxanthin	Regulates expression of TNF-*α*, CD80, CD86 and MHC II in RAW 264.7 cells.	[163]
*Penicillium* sp.	Brevicompanine E	Reduces the production of proinflammatory cytokines induced by LPS.	[164]
*Toxicocladosporium* sp. SF-5699.	Citreohybridonol	Suppresses neuroinflammatory enzymes and cytokines associated with NF-кB and MAPK in BV2 cells stimulated by LPS.	[184]
*Aspergillus insulicola*	Azonazine (Dipeptide)	Inhibits the production of NF-*κ*B luciferase and nitrite.	[165]
*Aspergillus* sp. SF-5921	Aurantiamide acetate	Exhibits NF-*κ*B, JNK, and p38 inhibition in BV2 microglia cells.	[185]
*Ascomycota* sp. CYSK-4	Isocoumarins	Inhibits the production of NO in LPS-induced RAW 264.7 cells	[166]
*Xylaria* sp. 2508	Xyloketal	Exhibits neuroprotective effect on neonatal hypoxic-ischemic brain injury both in vivo and in vitro.	[186]
*Eurotium amstelodami*	Questinol (Anthraquinone)	Inhibits NO and PGE2 production in LPS-stimulated RAW 264.7 cells.	[167]
*Eurotium* sp. SF-5989	Neoechinulins A and B (Diketopiperazine)	PGE2 and NO generation as well as iNOS and COX2 expression are downregulated. Diminishes IL-1 and TNF-*α* secretion.	[187]
*Chaetomium globosum*	Chaetoglobosin Fex	Suppresses LPS-stimulated IL-6, monocyte chemotactic protein-1, and TNF-*α* in peritoneal macrophages and mouse macrophage cells.	[188]
*Penicillium paxilli Ma(G)K*	Pyrenocine A	Inhibits gene expression in LPS-stimulated macrophages due to NF-*κ*B-mediated signal transduction.	[189]
*Ecklonia* *stolonifera*	Phlorofucofuroeckol (Phlorotannin)	Inhibits NO and PGE2 production by the suppressing iNOS and COX-2 protein expression.	[190]

CD, cluster of differentiation; COX, cyclooxygenase; IL, interleukin; JNK, c-Jun NH2-terminal kinase; iNOS, inducible nitric oxide synthase; LKB4; leukotriene B4; MAPK; mitogen-activated protein kinase; MHC, major histocompatibility complex; NF-kB, nuclear factor-*κ*B; NO, nitric oxide; PGE2, prostaglandin E2; PLA2, phospholipase A2; TLR, toll-like receptor; TNF, tumor necrosis factor.

**Table 6 marinedrugs-17-00282-t006:** Mangroves, corals and other marine creatures and their therapeutic chemical constituents.

Species	Chemical(s)	Immunomodulatory Activity	Ref(s)
*Ecteinascidia turbinate*	Yondelis (Trabectedin)	Reduces the proliferation of monocytes and the differentiation of *ex vivo* macrophages.	[191]
*Rhizophora apiculata*	Leaf extract	Inhibits HIV-1 or HIV-2 and reduces acute inflammation.	[174,175]
*Acorus calamus*	Rhizome extract	Inhibits cell proliferation and IL-2, NO, and TNF-*α* production is encouraged.	[176]
*Pseudopterogorgia elisabethae*	Diterpene glycosides	Inhibits TPA induced oedema in mouse, MPO release in human PMNs and, NO production in J774 macrophages.	[178]
*Stichodactyla helianthus*	Peptide ShK	Regulates the function of effector-memory T cells and class-switched memory B cells.	[179]
*Mytilus coruscus*	D-Glucan	Suppresses the production of LPS-induced TNF-*α*, NO, and PEG2.	
*Halocynthia aurantium*	Fatty acid	Increases production of NO and PGE2 in RAW 264.7 cells.	[182]
*Lepeophtheirus salmonis*	Trypsins	Causes an inhibitory effect on central inflammatory gene (*IL-1β*)	[192]
*Litopenaeus vannamei*	Polysaccharides	Exhibits immunomodulatory action of superoxide dismutase and its possible use as an indicator of immune responses.	[193]
*Nematopaleamon tenuipes* *Hemifusus pugilinus* *Euchelus asper* & *Rastrelliger kanagurta*	Fractions of Petroleum ether:ethyl acetate (1:1)	Exhibits immunosuppressive activity in the plaque forming cell assay.	[180,194]
*Crenomytilus grayanus*	Mytilan (Bioglycan)	Mytilan isolated from the mussel mantle *Crenomytilus grayanus* is highly immunomodulating.	[195]
*Bryozoans*	Convolutamydine A (Oxindole alkaloid)	Inhibits COX-2, iNOS, IL-6, PGE2 and TNF-*α* production.	[196]
*Seleronephthya gracillimum*	Pregnane-type steroids (Sclerosteroid)	Inhibits the expression of both iNOS and COX-2 proteins in LPS induced macrophages.	[197]
*Marthasterias glacialis*	Ergosta-7,22-dien-3-ol	Anti-inflammatory. Effective against iNOS, CHOP and I*κ*B-*α* expression.	[198]
*Astropecten polyacanthus*	Steroids	Inhibits pro-inflammatory cytokine secretion, including IL-12, p40, IL-6 and TNF-*α*.	[199]
*Lobophytum micchaelae*	Michosterols (Polyoxygenated steroids)	Suppresses the generation of superoxide anion and elastase release in human neutrophils stimulated by N-formyl-methionyl-leucyl-phenylalanine /cytochaslasine B.	[200]
*Paralemnalia thyrsoides*	Isoparalemmone (Sesquiterpenoid)	Inhibits iNOS protein expression in activated RAW 264.7 cells.	[201]
*Cladiella hirsuta*	Hirsutalins (Diterpenes)	Inhibits LPS-stimulated iNOS protein production.	[202]
*Lobophytum leavigatum*	Laevigatol	Inhibitory effects on NF-*κ*B-induced transcriptional activity in Hep-G2 cells.	[203]
*Sinularia gibberosa*	Gibberoketosterol (Steroids)	Inhibits the production of iNOS and COX-2 proteins in LPS-stimulated RWA 264.7 cells.	[204]
*Pseudopterogorgia elisabethae*	Pseudopterosins (Diterpene glycosides)	Blocks zymosan-induced eicosanoid release in RAW 264.7 cells.	[178]
*Eunicea fusca*	Fucosides (Diterpene arabinose glycosides)	Inhibits inflammation in the oedema model induced by 12-O- tetradecanoylphorbol-13-acetate.	[205,206]
*Hexaplex trunculus, Charonia tritonis*	Flesh and ashes of burned shell	Strengthens body’s immune system; sore and wound healing property.	[207]
*Potamididae*	Shell and flesh	Inhibits the inflammation of the mouth, recurrent aphthous ulcer, and gingivitis.	[208]
*Eudistoma toealensis*	Staurosporine & Enzastaurin	Ameliorates neuroinflammation by reducing demyelination and axonal damage.	[31]
*Haliotis discus hannai*	Extracts fermented with C. militaris mycelia (HFCM-5)	Inhibits the production of NO in RAW 264.7 cells.	[209]
*Capnella imbricate*	Capnellene	Inhibits iNOS and COX-2 in IFN-*γ*-stimulated microglial cells.	[210]
*Haliotis diversicolor*	Shell powder	Decreases iNOS expression and enhances the function of macrophages.	[211]
*Filopaludina bengalensis*	Footpad lipid extract	Inhibits ROS, TNF-*α*, and NO production.	[212]
*Dicathais orbita Gmelin*	Chloroform extract of the hypobranchial gland	Inhibits the production of NO, downregulated the production of TNFα in RAW 264.7.	[213]
*Perna canaliculus* *Gmelin*	Novel omega 3 polyunsaturated fatty acids	Inhibits the biosynthesis of cholesterol, COX-2, TNF-α and PGE. Inhibits TNF-*α* and IL-12p40 production in THP-I.	[214]
*Anadara kagoshimensis*	Polypeptide fraction	Inhibits NO in LPS-stimulated macrophage RAW 264.7cells. Inhibit IL-6, TNF-*α*, and IL-8 in human cervical cancer HeLa cells.	[215]
*Fissurella* *Latimarginata Sowerby*	Hemocyanin	Increases IFN-*γ* and higher numbers of tumor-infiltrating CD4+ lymphocytes. The generation of IL-6, IL-12, IL-23 and TNF-*α* in dendritic cells increases rapidly.	[216]
*Perna canaliculu Mytilus unguiculatus s,*	Lipid extract	Reduces the swelling of paw oedema. Inflammatory mediators (LTB4, PGE 2, and TXB2) and pro-inflammatory cytokines (IL-1, IL-6, INF-*γ*, and TNF-*α*) have been suppressed.	[217]
*Sepiella inermis*	Zhikang Capsule	Suppresses TNF-*α*, IFN-*γ*, IL-1*β*, and IL-12. Anti-inflammatory mediators (IL-4 and IL-10) have been promoted.	[218]
*Oily fishes*	Marine n-3 polyunsaturated fatty acids	Decreases human T cell spread, slows onset of arthritis, reduces paw swelling, reduces knee joint pathology, modulates a range of immunological reactions associated with RA.	[219]
*Sinularia kavarattiensis*	Sinuleptolide	IL-1*β*, IL-6, IL-8, IL-18, and TNF-α inhibition.	[220]
*Carijoa* sp.	Steroid glycoside carijoside	Neutrophil superoxide and elastase inhibition.	[221]
*Sinularia gyrosa*	Terpene gyrosanolides B & C	Inhibits iNOS expression in macrophages.	[222]
*Sinularia flexibilis*	11-Dehydrosinulariolide	Attenuates 6-OHDA-induced downregulation of TH-immunoreactivity and 6-OHDA-induced upregulation of DJ-1 protein in rat and zebrafish models.	[223]
*Klyxum simplex*	Klysimplexin sulfoxide (Terpene) & simplexin E	Inhibits expression of COX-2 and iNOS in macrophages.	[224,225]
*Lobophytum crassum*	Diterpenes	Inhibits NO release and iNOS expression in macrophages.	[226]
*Nephthea chabroli*	Nebrosteroid I (Steroid)	Inhibits iNOS expression in macrophages.	[227]
*Hyriopsis cumingii lea*	Polysaccharide	Activates adaptive immune response including T and B cells.	[227]
*Styela plicata*	Dermatan sulfate (Polysaccharide)	Lymphocyte and macrophage, as well as TNF-*α*, TGF-*β* and VEGF, have significantly decreased in inflamed colon of the rats.	[228]
*Lobophytum durum*	Durumhemiketalolide (Terpene)	Inhibits expression of macrophage COX-2 and iNOS.	[229]
*Lemnalia cervicorni*	Lemnalol	Inhibits spinal TNF-*α* in microglial cells and astrocytes in neurophathic rats.	[230]
*Sarcophyton ehrenbergi*	Glycolipid & sarcoehrenosides	Inhibits iNOS expression in macrophages.	[229]
*Sarcophyton crassocaule*	Sarcocrassocolides A & B (Terpene)	Inhibits iNOS expression in macrophages.	[231]
*Aplidium species*	Rossinones A & B (Terpene)	Inhibits neutrophil superoxide.	[232]
*Nephthea erecta* & *Nephthea chabroli*	Chabrosterol (Steroid)	Inhibits iNOS and COX-2 expression in macrophages.	[233]
*Mastigias papua*	Symbiopolyol (Polyketide sulfate)	Inhibited expression of inducible vascular cell adhesion molecule-1, which binds to leukocytes in early inflammation stages.	[234]
Shellfish & finfish sp.	Docosahexaenoic acid	Inhibits carrageenan-induced microglial activation, p38 MAPK phosphorylation, and *TNF- α* and *IL-1β* mRNA expression in spinal cord.	[235]

CHOP, C/EBP homologous protein; COX, cyclooxygenase; HIV, human immunodeficiency virus; IgE, immunoglobulin E; IL, interleukin; iNOS, inducible nitric oxide synthase; LKB4; leukotriene B4; MAPK; mitogen-activated protein kinase; MHC, major histocompatibility complex; NF-*k*B, nuclear factor-*κ*B; NO, nitric oxide; PGE2, prostaglandin E2; PMNs, polymorphonuclear neutrophils; PLA2, phospholipase A; TLR, toll-like receptor; TNF, tumor necrosis factor; TPA, 12-O-tetradecanoylphorbol13-acetate; TXB2, thromboxane B2.

## 3. Anti-inflammatory and Immunomodulatory Effects of the Chemical Constituents of Marine Flora 

Marine organisms are not only adapted to life in water with high salt concentrations, but have incorporated halogens into their chemical constituents, since ocean water contains chloride, bromide and iodide [194]. The extensive utilization of halogen ions by various marine organisms has important consequences for their overall composition. A plethora of chemical compounds have been discovered from this source. The chemical uniqueness of marine organism-derived compounds has accelerated drug discovery from those marine sources which have the highest probability of having novel molecules and interesting biological activity [236]. Marine flora is a prolific source of bioactive constituents including polysaccharides, oligosaccharides, terpenoids, steroids, alkaloids, polyphenols and antioxidants.

### 3.1. Polysaccharides

The most abundant and chemically complex organic molecules in the oceans are polysaccharides (Figure 1, **19**) [237]. Polysaccharides (Figure 1, **19**) of marine origin are a class of biochemical compounds that has been shown to have valuable therapeutic properties. These compounds are considered to be biocompatible and to have little or no toxicity. Marine algae and bacteria possess an extensive and valuable chemical library of unique polysaccharides. Sulphated polysaccharides can enhance the innate immune response by promoting the tumoricidal activities of macrophages and natural killer cells [136,238,239,240]. Polysaccharides (Figure 1, **19**) from macro- and microalgae have anti-inflammatory and immunomodulatory properties [241,242]. Among the marine algal polysaccharides, fucoidans, which are fucose-containing sulphated polysaccharides from brown seaweeds, have immunomodulatory effects [243,244]. EPCP1-2, a marine EPS extracted from *Crypthecodinium cohnii, e*xhibits anti-inflammatory activity achieved by regulating the TLR4 pathway [245]. P-KG03 sulphated polysaccharide, derived from marine microalgae *Gyrodinium impudium* strain KG03, activates the production of NO in a JNK-dependent manner and stimulates the production of cytokines IL-1*β* and 6 and TNF-*α* in macrophages, and prevents the growth of tumor cells in vitro and in vivo [136,150]. Alginic acid, a colloidal polysaccharide from brown seaweed, inhibits the secretion of TNF-*α* and IL-1 [246]. Several bacteria found in the deep-sea, in shallow hydrothermal vents, the Antarctic, and hypersaline lakes produce EPS [247,248,249,250,251]. EPS1 from a haloalkaliphilic, thermophilic strain of *Bacillus licheniformis* T14 hinders HSV-2 replication in hPBMCs. The non-cytotoxic exopolymer EPS1-T14 can stimulate the immune response and thus contribute to host defense against viruses [23].

### 3.2. Alkaloids

Alkaloids, a structurally diverse group of secondary metabolites containing nitrogen, have a range of biological activities. Alkaloids are mostly found in higher plants, but many marine organisms also contain alkaloids [252,253]. Alkaloids from the marine sponges *Axinella verrucosa* and *Acanthella aurantiaca* have been characterized as NF-*κ*B-specific inhibitors [100]. An oxindole alkaloid, convolutamydine A (Figure 1, **20**) and its two analogs, ISA003 and ISA147 from marine bryozoans inhibits the formalin-induced licking behavior significantly in mice, migration of leucocytes, and expression of COX-2, PGE2, iNOS, IL-6, and TNF-*α* in RAW 264.7 cells [196]. Neoechinulins A (Figure 1, **21**) and B, two diketopiperazine indole alkaloids from marine fungus *Eurotium* sp. SF-5989 exert in vitro anti-inflammatory activity on LPS-stimulated RAW 264.7 cells. Neoechinulin A (Figure 1, **21**) was considered safe in vitro using a cell viability assay, but neoechinulin B exhibited toxicity. Neoechinulin A (Figure 1, **21**) derived from *Microsporum* sp. downregulates the formation or expression of COX-2, PGE2, NO, ROS, iNOS, IL-1, IL-6, and TNF-*α* in oligomeric amyloid-*β* activated BV-2 microglial cells. This compound also downregulates apoptosis mediated by activated microglia in pheochromocytoma PC-12 cells and reduces the nuclear translocation of NF-*κ*B p50 and p56 subunits. Neoechinulin A (Figure 1, **21**) can also inhibit neuroinflammation in Alzheimer’s disease [254]. Cytochalasan-based alkaloid chaetoglobosin Fex (Cha Fex) (Figure 1, **22**), isolated from the fungus *Chaetomium globosum*, suppresses IL-6, TNF-*α* and monocyte chemotactic protein-1 in LPS-stimulated peritoneal macrophages and RAW 264.7 cells. Cytokine mRNA expression is lowered, entry of the p65 subunit of NF-*κ*B into the nucleus and LPS-elicited breakdown of I*κ*B*α* is impaired, and the levels of the extracellular-signal-related kinase (ERK1/2), p38, and c-Jun is reduced by Cha Fex (Figure 1, **22**) alkaloid. In addition, the upregulation of membrane-associated CD-14 expression induced by LPS on RAW 264.7 cells and human monocytes was suppressed [188]. Alkaloids have been used by humans for a variety of purposes for more than 4000 years. Alkaloids and alkaloid-containing taxa will undoubtedly continue to play an important role in modern drug development [255].

### 3.3. Polyphenols

More than 8000 polyphenolic compounds are found in marine flora, including phlorotannins, flavonoids, anthocyanins, tannins, lignin, epigallocatechin, epicatechin, catechin, and hydroxylated polybrominated diphenyl ethers [256,257,258]. Polyphenols with multiple phenolic structural units are bioactive and are widely distributed in plants [257] and have a wide range of biological activities including antioxidant [259], cardiovascular protective, anti-cancer [260,261], anti-inflammatory and immune-modulatory effects [262,263]. Modern molecular and cellular biology techniques have led to a greater understanding of the benefits arising from polyphenols [264,265,266,267]. Cellular signaling and regulation of gene expression by polyphenols through modulation of NF-*κ*B has a significant impact on cancer and chronic inflammation. For instance, resveratrol acts on the NF-*κ*B pathway at multiple levels and is able to down-regulate its expression, phosphorylation and transcription activity [268,269,270,271]. Diphlorethohydroxycarmalol (DPHC) (Figure 1, **23**), a phlorotannin from *Ishige okamuarae,* exerts an anti-inflammatory effect by strongly inhibiting IL-6 production in LPS-stimulated RAW264.7 cells. In addition, DPHC inhibits the expression of signal transducer and activator of transcription 5 (STAT5) signaling and increases the production of suppressor of cytokine signaling 1 (SOCS1) [151]. A phlorotannin sub-fraction isolated from *Fucus distichus* reduces TNF-*α*, IL-10, MCP-1 and COX-2 expression. The sub-fraction also lowers downstream TLR activation and expression of inflammatory biomarkers. In view of the potential cellular signaling capabilities of polyphenols, phlorotannin is beneficial because it acts as a free radical scavenger, and can also modulate inflammatory signaling receptors such as TLRs and downstream protein pathways, including NF-*κ*B, JNK and p38 MAPKs [152]. 

### 3.4. Steroids/Sterols

Steroids are lipophilic compounds derived from cholesterol, and have a variety of marine, terrestrial, and synthetic sources. Steroids and their metabolites play an important role in the physiology and biochemistry of living organisms. For example they are used as hormone antagonists [272], contraceptives [273], cardiovascular therapeutic agents [274], anti-cancer agents [275], osteoporosis treatments [276], anesthetics, antibiotics, anti-asthmatics, and anti-inflammatories [277]. 

Steroidal compounds isolated from sponges that modulate the pregnane X receptors (PXRs) are effective in reducing intestinal inflammation by manipulating NF-*κ*B activity [278,279]. Solomonsterol A (Figure 1, **24**) from the marine sponge *Theonella swinhoei* is a selective PXR agonist and has been shown to mitigate systemic inflammation and immune system disturbances in a RA mouse model. Solomonsterol A (Figure 1, **24**) inhibits the development of arthritis caused by anti-collagen antibodies in transgenic mice harboring a humanized PXR2. Solomonsterol A (Figure 1, **24**) reduces the expression of the inflammatory markers TNF-*α*, IFN-*γ* and IL-17 and chemokines MIP1-*α* and RANTES, which reduces the inflammatory response [280]. Pregnane-type steroids derived from the soft coral *Seleronephthya gracillimum* show in vitro anti-inflammatory activity by reducing the accumulation of the pro-inflammatory proteins iNOS and COX-2. Sclerosteroid K (Figure 1, **25**) and M (Figure 1, **27**) reduce iNOS accumulation, while LPS-stimulated COX-2 accumulation in RAW 264.7 macrophages is prevented by these sclerosteriods [197]. Ergosta-7, 22-dien-3-ol (Figure 1, **28**), an unsaturated sterol from the spiny sea star *Marthasterias glacialis* has anti-inflammatory activity on RAW 264.7 cells in vitro. The inflammatory markers NF-*κ*B, iNOS, IL-6, and COX-2 are downregulated. Sterol ergosta-7, 22-dien-3-ol is most effective, but a potentially synergistic effect was obtained when this compound was administered with other compounds [198]. Steroids isolated from the starfish *Astropecten polyacanthus,* downregulate the generation of IL-6, IL-12 p40, and TNF-*α* in LPS-stimulated bone marrow-derived dendritic cells [199]. Polyoxygenated steroids michosterols A-C extracted from the soft coral *Lobophytum micchaelae* exhibit potent anti-inflammatory activity in stimulated neutrophils by suppressing the superoxide anion generation and elastase release in N-formyl-methionyl-leucyl-phenylalanine/cytochaslasine B *(*fMLP/CB) [200].

### 3.5. Miscellaneous Compounds with Anti-oxidant Activities

Antioxidants are agents which modulate the levels of highly reactive oxygen species (ROS), which cause damage by binding to biomolecules such as DNA. Antioxidants act by neutralizing ROS produced during biochemical reactions. In diseases such as Alzheimer’s, Parkinson’s, atherosclerosis, stroke, cancer, diabetes, RA, and IBD the antioxidant potential of bio-modulators has been studied and the underlying prophylactic and therapeutic aspects investigated [281,282,283,284,285,286]. Antioxidants can function as immune modulators and are used in conjunction with mainstream therapy in some diseases. Antioxidants have been used in disease prevention, where they serve as free radical scavengers. Tocopherols in humans and mice suppress PGE2 synthesis and enhance cell-mediated immunity [287,288,289]. Selenium augments the phagocytic abilities of macrophages and prevents CD8+ T lymphocyte damage [290]. Astaxanthin and fucoxanthin, marine carotenoids, appear to be biologically more effective than terrestrial carotenoids [291,292,293,294,295]. Astaxanthin decreases the production of NO, iNOS activity, and the production of PGE2 and TNF-*α* in RAW 264.7 cells in a dose-dependent manner [296]. Fucoxanthin prevents inflammation by inhibiting inflammatory cytokines TNF-*α* and IL-1, and limits the expression of COX-2 and iNOS, as well as eliminating excess ROS [297]. 

Vitamin C is an electron donor and acts as potent water-soluble antioxidant helping to prevent protein, lipid and DNA oxidation. Supplementation with vitamin C reduces IgE and histamine levels by increasing IFN-*γ* and deceasing IL-4, an observation which indicates the suppression of immune response of type Th2-type cytokines [298]. Trace elements such as iron, copper, selenium and antioxidants are found in eight species of red (*Hypnea spinella, Gracilaria textorii, Gracilaria vermicullophyla),* green (*Caulerpa sertularioides, Codium simulans, Codium amplivesiculatum Ulva lactuca*) and brown (*Dictyota flabellata*) microalgae [299]. Due to environmental conditions such as temperature and high levels of irradiation, microalgae and seaweeds have high levels of ROS, which seaweeds deactivate with their high intracellular amounts of antioxidant compounds such as polyphenols, phycobilins, carotenoids and vitamins [300].

## 4. Metabolic Engineering and Genomic Approaches for Marine Compounds

Marine compounds that are used as drugs in microorganisms, plants and animals are synthesized in small amounts and are difficult to obtain in large quantities. This is where metabolic engineering comes into play. Using metabolic engineering, we can model organisms using existing metabolic reconstructions to discover gene knockouts which could improve the yield of products of interest. Genome-scale metabolic constraint-based flux models have been constructed for *Streptomyces coelicolor* A3 strains with the aim of improving and optimizing the production of antibiotics [301]. The deletion of the *gonCP* gene in marine actinobacterium *Streptomyces coniferous* resulted in improved antitumor activity of two derivatives, PM100117 and PM100118 [302]. By combining metabolic engineering and mutagenesis, it was possible to produce a level of astaxanthin in *Xanthophyllomyces dendrorhous* that was 89 times higher than that of the wild-type strain [303]. Advances in the understanding of microbial metabolic pathways, together with the use of new combinatorial techniques and random mutagenesis, can increase product yield and enhance efficacy [304]. Genome-scale metabolic models have been developed for several non-marine organisms and several are currently used in industrial settings. Using genomic proteomic transcriptomic and other omics data across various conditions from in-vivo experiments and literature of marine organisms the genome-scale metabolic model can be developed to produce overproducing strains of a target organism (Figure 2) [305]. In many cases, the chemical structures of biomolecules can be predicted to a certain extent, based on the analysis and biosynthetic logic of the enzymes encoded in a biosynthetic gene cluster, and their similarity to known counterparts [306]. Genomic analysis provides new insights into marine biodiversity and can reveal new drug sources. Retrieval of genomic information from marine microorganisms can also be used for the discovery of new drug molecules from microorganisms that are yet to be cultivated [307]. The CRISPR-Cas9 gene editing technology has been used in large number of microorganisms for genome modulation [308]. Researchers assembled a multifunctional metabolic engineering system based on CRISPR, which makes use of an RNA-guided nuclease. This system allows metabolic engineers to modify microorganisms via gene modifications and substitutions, and provide a practical means to reduce metabolic flux through redundant metabolic pathways and direct energy towards production of the target compound [309].

## 5. Conclusions

Chemical substances derived from marine organisms have proven to be a very effective in the prevention and treatment of disease. The recent development of new marine-derived treatments for cancer, inflammation and infectious diseases suggest that a focus on the development of marine medicines could be very valuable. The discovery of novel chemicals with therapeutic potential from marine sources requires the exploration of unique habitats such as deep-sea environments, as well as isolation and culturing of marine microorganisms. The traditional pharmaceutical fit of marine microorganisms as model natural product drug sources makes them more attractive. The production of bulk quantities of microbe-derived drugs is still challenging and can be addressed using metabolic engineering, in order to meet the growing need for a wide range of pharmaceuticals. Marine organisms both known and as yet undiscovered, may hold answers to some of our most pressing medical problems. 

## Figures and Tables

**Figure 1 marinedrugs-17-00282-f001:**
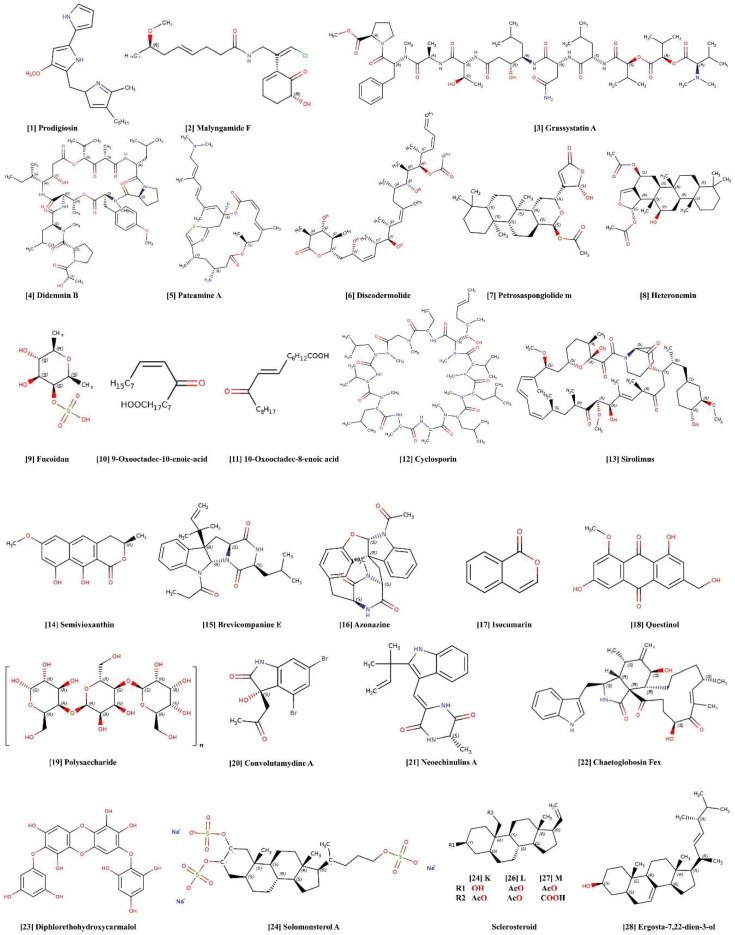
Structure of anti-inflammatory and immunomodulatory marine-derived compounds.

**Figure 2 marinedrugs-17-00282-f002:**
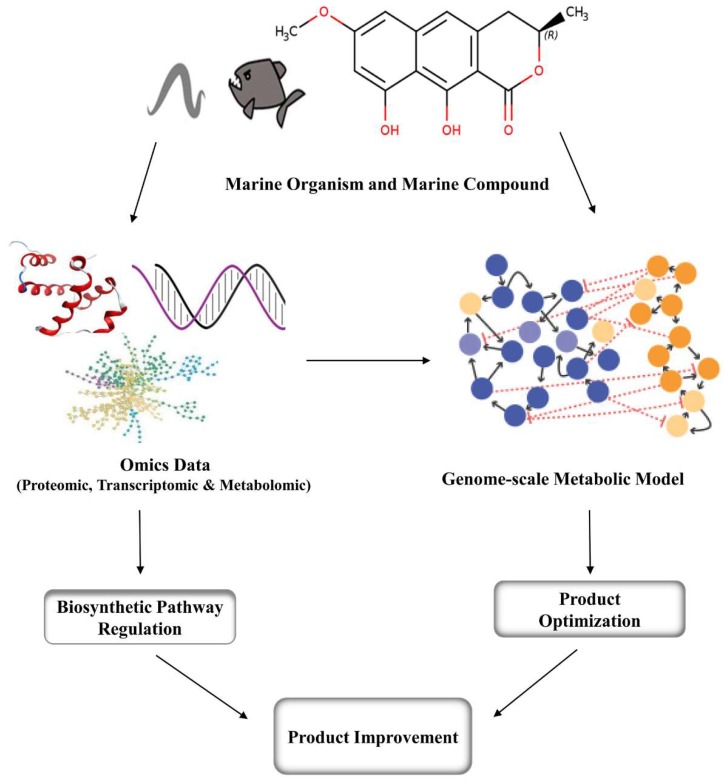
Schematic diagram showing omics data analysis and genome-scale metabolic modelling for improvement of production.

**Table 1 marinedrugs-17-00282-t001:** Marine bacteria and their therapeutic chemical constituents.

Bacterial Species	Chemical(s)	Immunomodulatory Activity	Ref(s)
*Aplidium albicus*	Cyclic depsipeptide	In vivo activity in the plasmocytoma murine model of xenograft. Applidine exhibits antimyeloma activity in vivo.	[30]
*Nocardiopsis* sp. K-252, *Nonomuraea longicatena*	Lestaurtinib (Alkaloid)	Potent PKC and calmodulin inhibitor. Prevents myelin oligoglycoprotein induced encephalomyelitis in vivo.	[31]
*Bryozoa neritina*	Bryostatin polyketide	Exhibits antitumor activity against malignant melanoma. IL-6 and TNF-*α* levels rise in patients within 24 hours of the treatment.	[32,33]
*Micrococcus luteus*	Anti-Micrococcus luteus antibodies	Immunosuppressive potential through the expansion of immunoregulatory T cell subsets.	[34]
*Trididemnum solidum*	Didemnin B (Depsipeptides)	Exhibits strong anti-inflammatory and immunosuppressive activity. The expression of iNOS and NF-*κ*B was inhibited in vitro.	[35,36]
*Bacillus licheniformis*	EPS 1-T14	Stimulates Th1 cell-mediated immunity.	[23,24]
*Thermus aquaticus*	EPS TA-1	Encourages the TLR2-dependent release of TNF-*α* and IL-6 in murine macrophages.	[24]
*Serratia marcescens, Vibrio psychroerythrus, Pseudoalteromonas denitrificans & Zooshikella rubidus*	Prodigiosin & Cycloprodigisin	Anti-inflammatory. Inhibits the activation of TNF-*α* induced NF-*κ*B.	[27,28,37]
*Bacillus* sp. HC001, *Piscicoccus* sp. 12L081	Diketopiperazines	Anti-inflammatory. Downregulates the release of TNF-*α* and IL-6 and suppress NF-*κ*B expression.	[22]
*Salinispora arenicola*	Arenamides (Cyclic depsipeptide)	Blocks TNF-*α* in RAW 264.7 and human embryonic kidney cells.	[38,39]
*Streptomyces* sp. SCRC-A20	Aburatubolactams	Antioxidant. Inhibits TPA-induced superoxide anion generation in human neutrophils.	[40]
*Streptomyces* sp. CNB-982	Cyclomarins (Heptapeptides)	Anti-inflammatory. Inhibits oedema and pain in vivo.	[41]
*Streptomyces* sp.	Salinamides (Peptides)	Anti-inflammatory on phorbol ester-induced oedema mouse.	[42]
*Streptomyces strain* CNQ43	Splenocin B	Anti-inflammatory. Potent inhibitors of pro-inflammatory cytokine IL-5, IL-13 and TNF-*α*.	[43]

EPS, exopolysaccharide; iNOS, inducible nitric oxide synthase; IL, interleukin; NF-*κ*B nuclear factor-*κ*B; PKC, protein kinase C; TPA, 12-O-tetradecanoylphorbol-13-acetate; TLR, toll-like receptor; TNF, tumor necrosis factor.

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
