# Peer review of "Oceans as a Source of Immunotherapy"

_marinedrugs, 2019, doi:10.3390/md17050282_

Reviewer 1 Report

The work entitled “Oceans as a source of new immunotherapy” includes an extense review regarding different molecules with marine origin that have proved to have potential as immunomodulators. While most os the references are relatively modern, some of them are quite old. The knowledge regarding the potential of marine originated drugs has been known since a long time ago, so the title of the review and the conclusions where words like new, novel, recent development, are not that appropriate.

It can be appreciated a hard revision work and the organization of the review is quite good, however I miss some discussion of the bibliography found, since sometimes some moleculaes have proved opposite effects. I also miss a section including the compounds that have already been approved and are available on the market, like Yondelis. 

I accept after minor revision (corrections to minor methodological errors and text editing). Some general issues must be adressed: 

- Authors should include a discussion in those cases that a molecule has proved opposite effects.

- TNF is Tumor Necrosis Factor, not Necrotic. This should be corrected in all cases.

- Authors should use always the same name for this cell type: RAW 264.7 or Raw 264.7.

- Authors should carry out an extense and exhaustive revision of the Bibliography section. There are many mistakes, some of them are described below:

        - In the following reference the journal name does not appear and the title of the article is written in italics: 30, 67, 69, 87,134, 170, 175, 184, 212, 230, 235, 264, 309

        - Revise reference 71, no authors are included.

- The text included in the immunomodulatory activity section in all the tables should be homogeneus. In some cases, authors write a short description of the immunomodulatory effect, while another a longer one. And sometimes there are inconsistencies in verb times (mixture of present, present perfect…). It seems that it has been copied directly from the original article. In addition, the cell type, the animal over which the drug has effect, and if the study has been performed in vivo or in vitro, should also be included in the tables. This would increase al lot the quality of the review.

- Some scientific names are not in italics. Please revise.

- All the tables must be eddited. 

Some details must be adressed too:

- In line 58, authors talk about marketed formulations but not for what purpose. The reference refers to antitumoral. This should be included.

- I would suggest, when refering to structures defined in figure 1, citate figure 1 in the main text in addition to the number that represents each structure.

- In table 3, regarding marine sponges, authors have included Ulva fasciata that is an Algae. In addition for this especies and the one before (Dendrilla nigra), the lipopolysaccharide appears as the chemical, is that right?

- In table 3: Discodermia especies and Reniera species, shouldn´t be Discodermia spp and Reniera spp?

- The text in lines from 240-276 is duplicated.

- Revise the text corresponding to the citation 178 in line 257: “express cell-surface markers CD16”. The other markers described in the original article should be included (HLA-DR, and also CD25 is downregulated).

- Authors should revise the order of the citations. In line 271 the reference 183 is citated and the following one in line 275 is the 193, not consecutive…

- I can not find reference 175 on the internet. Authors should take care when obtaining references from a review, they are not always well referenciated. This should be revised.

- In table 6, the last citation refers to a fish but it does not include the species. This should be included.

- In lines 381-382, authors describe the pro-inflammatory potential of polyphenols, but then citate organisms with anti-inflammatory effect. Is this right?

- The 3.5 section describes antioxidant compuds that is a type of compound but not a type of biomolecule. This different clasiffication herein seems extrange.

Author Response

Reviews No 1.

The work entitled “Oceans as a source of new immunotherapy” includes an extensive review regarding different molecules with marine origin that have proved to have potential as immunomodulators. While most of the references are relatively modern, some of them are quite old. The knowledge regarding the potential of marine originated drugs has been known since a long time ago, so the title of the review and the conclusions where words like new, novel, recent development, are not that appropriate.

Response: We appreciate the referee for considering our work worth publishing in esteemed journal “Marine Drugs”. As the data of marine originated drugs has been produced since ages, we have tried to compile old as well as new data into this study. Favorably considering the referee’s suggestion, we have modified the title of the study as: Oceans as a source of immunotherapy”.

It can be appreciated a hard revision work and the organization of the review is quite good, however I miss some discussion of the bibliography found, since sometimes some molecules have proved opposite effects. I also miss a section including the compounds that have already been approved and are available on the market, like Yondelis.

I accept after minor revision (corrections to minor methodological errors and text editing). Some general issues must be addressed:

Comment: Authors should include a discussion in those cases that a molecule has proved opposite effects.

Response: With gratitude, the sole purpose of this study is to compile the data related to drugs of marine source with anticancer, immunomodulatory, and anti-inflammatory effects. Discussing the controversial effects as well as detail mechanism of every compound is beyond the scope of this study. We have tried our best to comprehend the data as much as possible. Providing further details will lengthen the text unnecessarily. This will change the main theme of the review.

Comment: TNF is Tumor Necrosis Factor, not Necrotic. This should be corrected in all cases.

Response: Correction has been made as per referee’s suggestion.

Comment: Authors should use always the same name for this cell type: RAW 264.7 or Raw 264.7.

Response: This and other related corrections have been made throughout the text.

Comment: Authors should carry out an extensive and exhaustive revision of the Bibliography section. There are many mistakes, some of them are described below:

In the following reference the journal name does not appear, and the title of the article is written in italics: 30, 67, 69, 87,134, 170, 175, 184, 212, 230, 235, 264, 309

Response: We appreciate referee’s suggestion and pointing these typos. We have tried our best to follow the journal’s guidelines for references formatting. All references have been updated and corrected wherever needed.

Comment: Revise reference 71, no authors are included.

Response: All references have been duly checked and corrected.

Comment: The text included in the immunomodulatory activity section in all the tables should be homogeneous. In some cases, authors write a short description of the immunomodulatory effect, while another a longer one. And sometimes there are inconsistencies in verb times (mixture of present, present perfect…). It seems that it has been copied directly from the original article. In addition, the cell type, the animal over which the drug has effect, and if the study has been performed in vivo or in vitro, should also be included in the tables. This would increase al lot the quality of the review.

Response: We thank the reviewer for this valuable suggestion, which has greatly enhanced the quality of the study. Taken consistency into consideration, we have edited the text and provided more elaborated description of the immunomodulatory activity section of the tables. In addition to this, the in vitro and in vivo data has been updated and included wherever was missing.

Comment: Some scientific names are not in italics. Please revise.

Response: We convey our thanks to the anonymous reviewer for the valuable query. The scientific names mention in the review has been revised as per scientific nomenclature.

Comment: All the tables must be edited. Some details must be addressed too:

Response: We have revised all the tables, and more details have been added.

Comment: In line 58, authors talk about marketed formulations but not for what purpose. The reference refers to antitumoral. This should be included.

Response: After detailed study of the references pointed by the referee, we found that the information given in line N. 58 of our unrevised manuscript is correct. Which says: “Over 60% of the active compounds of marketed formulations are natural products or their synthetic derivatives or mimics”.

The figure (60%) is describing the drugs of natural source with no define purpose (mode of action) provided in the given reference. Please refer to the figure 11 in the given link.

Comment: I would suggest, when referring to structures defined in figure 1, citate figure 1 in the main text in addition to the number that represents each structure.

Response: We thank the reviewer for this suggestion. We have now cited the figure 1 alongside structure’s number mentioned in the main text.

Comment: In table 3, regarding marine sponges, authors have included Ulva fasciata that is an Algae. In addition for this especies and the one before (Dendrilla nigra), the lipopolysaccharide appears as the chemical, is that right?

Response: Ulva fasciata is an algae and has been shifted to the related section. Lipopolysaccharide is an active chemical in these species with proven potent immunomodulatory effects.

Comment: In table 3: Discodermia especies and Reniera species, shouldn´t be Discodermia spp and Reniera spp?

Response: This correction has been made as per suggestion.

Comment: The text in lines from 240-276 is duplicated.

Response: Considering it as a human error, the duplicate text corresponding to line 240-276 has been deleted.

Comment: Revise the text corresponding to the citation 178 in line 257: “express cell-surface markers CD16”. The other markers described in the original article should be included (HLA-DR, and also CD25 is downregulated).

Response: Thanks for this valuable suggestion. We have revised the text corresponding to this citation. The revised text is given below (lines 268-269):

“In hPBMCs the production of IL-2, NO, and TNF-α was inhibited, IFN-γ and cell-surface markers CD16 and HLA-DR were not affected, but CD25 was downregulated”.

Comment: Authors should revise the order of the citations. In line 271 the reference 183 is citated and the following one in line 275 is the 193, not consecutive…

Response: We thank the reviewer for this query. We have updated this citation order while updating references.

Comment: I cannot find reference 175 on the internet. Authors should take care when obtaining references from a review, they are not always well referenciated. This should be revised.

Response: Thank you for the suggestion. We have tried our best to follow the journal guidelines and made sure each reference is up-to-date and accessible. The above mentioned refence could be accessed at the given link.

Comment: In table 6, the last citation refers to a fish but it does not include the species. This should be included.

Response: Docosahexaenoic acid (DHA) is found in more than 40 species of fishes including shellfishes and finfishes. Thus, we used a general fish term. However, shellfish and finfish sp have been mentioned in the revised manuscript.

Comment: In lines 381-382, authors describe the pro-inflammatory potential of polyphenols, but then citate organisms with anti-inflammatory effect. Is this right?

Response: In the revised manuscript we have removed the line which was misleading the referee. We did not mention the pro-inflammatory effect of polyphenols, rather the immune modulatory effect pf these compounds in tumor environment was mentioned. However, this confusion has been removed in the revised manuscript.

Comment: The 3.5 section describes antioxidant compounds that is a type of compound but not a type of biomolecule. This different classification herein seems strange.

Add comment

Thank you for the suggestion. The section name has been changed and revised heading is given below:

“3.5 Miscellaneous compounds with anti-oxidant activities”

Reviewer 2 Report

Bilal Ahmad et al. describe a series of marine molecules with immunomodulatory potential. Despite suggested so by the title, “…new immunotherapy”, this review presents only a collection of some fairly old, widely known and established concepts.

I cannot recommend to accept this paper. In order to be acceptable, it must be re-writtten focusing on recent developments. I also suggest to include a schemes into the review, such as for describing the different types of molecules and molecular mechanisms

Author Response

Reviewer No 2

Bilal Ahmad et al. describe a series of marine molecules with immunomodulatory potential. Despite suggested so by the title, “…new immunotherapy”, this review presents only a collection of some fairly old, widely known and established concepts.

I cannot recommend to accept this paper. In order to be acceptable, it must be re-writtten focusing on recent developments. I also suggest to include a schemes into the review, such as for describing the different types of molecules and molecular mechanisms.

Response: This study focuses on the marine-originated bioactive molecules with immunomodulatory potential that have been reported over that last five decades. The concerned referee has raised a point by including old studies while the title conveys a different message. Exploring the oceanic source for novel and new immune-modulatory drugs is a hectic job and it takes years to bring effective agents to the market or be evaluated in laboratories. The main purpose of this study is to collect immune modulating drugs of natural source, particularly ocean, into a single draft and facilitate the broad-range readers of the esteemed journal.

There are very few marine-based immune-modulating moleculs (enlisted and cited in revised version) reported in the last few years and restricting our study to the detail mechanism of those drugs with some mechanism will not benefit the broad-range readers, but a specific group of people having expertise in the field.

We have tried our best to update the information, by adding up-to-date references, and made changes in the draft, as much as we could. We have modified the title of the study as well, to make it consistent with the data presented in the main text.

We believe this will satisfy the referee’s concern.

Round  2

Reviewer 2 Report

The authors reviewed the manuscript and the recent bibliography was introduced.

The review can be accepted after the authors have checked all the molecules in Figure 1. 

For example in the first molecule the position of the double bonds is not correct as well as in the second structure the double bond in the cyclic structure is omitted. 

Author Response

The authors reviewed the manuscript and the recent bibliography was introduced.

Comment: The review can be accepted after the authors have checked all the molecules in Figure 1.

For example, in the first molecule the position of the double bonds is not correct as well as in the second structure the double bond in the cyclic structure is omitted. 

Answer: The authors are grateful for the timely review and valuable suggestions provided by the referee. The structures in figure 1 have been updated and provided in the revised manuscript. For referee’s convenience, please refer to the links below for each compound.

Compound 1, Compound 2, Compound 3, Compound 4, Compound 5, Compound 6, Compound 7, Compound 8, Compound 9, Compound 10, Compound 11, Compound 12, Compound 13, Compound 14, Compound 15, Compound 16, Compound 17, Compound 18, Compound 19, Compound 20, Compound 21, Compound 22, Compound 23, Compound 24, Compound 25, 26, 27, Compound 28.

We hope this will satisfy the referee’s concern.
